# Effects of Supplementation with Microalgae Extract from *Tetradesmus obliquus* Strain Mi175.B1.a on Gastrointestinal Symptoms and Mental Health in Healthy Adults: A Pilot Randomized, Double-Blind, Placebo-Controlled, Parallel-Arm Trial

**DOI:** 10.3390/nu17060960

**Published:** 2025-03-10

**Authors:** Sydnie Maltz, Aaron T. Nacey, Jonathan Maury, Nancy Ghanem, Sylvia Y. Lee, Thomas M. Aquilino, Elliot L. Graham, Scott D. Wrigley, Jennifer M. Whittington, Afsana M. Khandaker, Rania A. Hart, Lena Byrne, Yuren Wei, Rémi Pradelles, Sarah A. Johnson, Tiffany L. Weir

**Affiliations:** 1Department of Food Science and Human Nutrition, Colorado State University, Fort Collins, CO 80523, USA; sydnie.maltz@colostate.edu (S.M.); aaron.nacey@colostate.edu (A.T.N.); nancy.ghanem@colostate.edu (N.G.); sylvia.lee@colostate.edu (S.Y.L.); thomas.aquilino@colostate.edu (T.M.A.); elliot.graham@colostate.edu (E.L.G.); scott.wrigley@colostate.edu (S.D.W.); jenny.whittington@colostate.edu (J.M.W.); afsana.khandaker@colostate.edu (A.M.K.); lena.byrne@colostate.edu (L.B.); yuren.wei@colostate.edu (Y.W.); 2Research & Development Department, Microphyt, Mudaison, 34670 Baillargues, France; jonathan.maury@microphyt.eu (J.M.); remi.pradelles@microphyt.eu (R.P.)

**Keywords:** anxiety, blood pressure, gastrointestinal symptoms, gut microbiota, intestinal health, mental health, microalgae, microbiome, phytochemicals, sympathetic nervous system, *Tetradesmus obliquus*, stress

## Abstract

Microalgae, a marine-derived natural ingredient, has emerged as a rich source of bioactive compounds with the potential to modulate gut–brain axis activities. The objective of this study was to investigate whether supplementation with a microalgae extract from *Tetradesmus obliquus* strain Mi175.B1.a (TOME) influences gut health and reduces stress and anxiety in healthy adults experiencing mild to moderate gastrointestinal (GI) distress. **Methods:** Fifty-six healthy adults (age: 31.9 ± 7.7 years; body weight: 71.8 ± 12.6 kg; BMI: 24.6 ± 2.8 kg/m^2^) were enrolled in a randomized, double-blind, placebo-controlled, parallel-arm clinical trial. Participants were randomly allocated to receive capsules containing either 250 mg/day of TOME or a placebo for four weeks. Primary outcomes included the assessment of GI symptoms using the Gastrointestinal Symptom Rating Scale (GSRS) and Bristol Stool Scale (BSS). Secondary outcomes focused on subjective evaluation of mood, stress, and anxiety, as well as blood pressure responses to sympathetic nervous system activation induced by the cold pressor test (CPT). In addition, stool, plasma, and saliva samples were collected to assess biomarkers associated with stress, sympathetic activation, intestinal permeability, and GI health. 16S rRNA sequencing was performed to analyze changes in gut microbial populations. **Results:** Daily supplementation for four weeks with TOME was safe and well tolerated in the study population. In addition, TOME significantly reduced GSRS global scores (*p* = 0.02), as well as constipation (*p* = 0.05) and indigestion (*p* = 0.03) subcomponent scores compared to Placebo. There was also a significant increase in Shannon’s index before FDR correction (*p* = 0.05; FDR = 0.12) and stool butyrate level was significantly lower in the TOME group than in Placebo after 4 weeks of supplementation (*p* = 0.039). Both groups showed a significant reduction in perceived stress scores, but the TOME intervention group also had reduced Negative Affect scores (*p* < 0.001). In addition, plasma chromogranin A, a stress biomarker, was significantly reduced after TOME intervention (*p* = 0.03). There were no negative effects on blood lipids or other parameters related to sympathetic activation or cardiovascular health. **Conclusions:** Overall, these results suggest that 4-week supplementation with *T. obliquus* strain Mi175.B1.a improves GI symptoms, potentially through effects on the gut microbiota, and may promote positive effects on mental health. Additional research should follow up on mental health outcomes in populations with increased stress and anxiety and investigate mechanisms underlying improvements in GI health. This trial was registered at clinicaltrials.gov as NCT06425094.

## 1. Introduction

Gastrointestinal (GI) disorders and symptoms are highly prevalent, affecting over 40% of the global population, with a higher incidence among women and younger adults compared to those over 65 years of age [1]. Beyond their substantial economic burden on healthcare systems, GI disorders can adversely influence various physiological functions (e.g., cognitive, mental, immune, cardiovascular, and physical function), disease risk, and overall health-related quality of life [2,3,4,5].

Mental health, another major global health concern, has been increasingly linked to gut microbiota and the nervous system through the gut–brain axis (GBA) [4,6,7]. This bidirectional communication system connects the gut and the central nervous system via neural, hormonal, immune, and neurotransmitter signaling pathways [7]. Alterations in the gut microbiota and their metabolites are associated with the regulation of neuroinflammation and sympathetic nervous system-mediated blood pressure regulation, linking the gut microbiota and neuroinflammation to hypertension [8,9,10]. The mechanisms to support this hypothesis are still limited, but some evidence suggests that gut dysbiosis results in an imbalance of circulating anti- and pro-inflammatory mediators of host systemic inflammation. Dysbiosis-induced inflammation may directly and indirectly activate the microglia to induce neuroinflammation, a heightened immune system activation [7,9,11].

Disruptions in gut health can also lead to increased intestinal permeability. As a result, metabolites like lipopolysaccharides (LPS) and other pathogen-derived metabolites decrease tight-junction integrity, allowing various pro-inflammatory compounds and endotoxins to translocate from the lumen, activate microglia, enter the bloodstream, and modulate immune responses, resulting in the genesis of pro-inflammatory cytokines [12]. Consequently, increased permeability of the epithelium coincides with increased permeability of the blood–brain barrier (BBB), since LPS and inflammatory cytokines also degrade the tight junction proteins of the BBB [12]. The increase in BBB permeability has been linked to observed increases in the development of mood disorders such as depression, as well as increased stress and anxiety. Additionally, gut microbiota-induced activation of the hypothalamic–pituitary–adrenal (HPA) axis increases the release of neuroendocrine stress markers like Chromogranin A, and reduces short-chain fatty acid (SCFA) production [7,13,14,15,16,17]. Chronic activation of the HPA axis and glucocorticoid hormone release are linked to increased blood pressure, hypertension, vascular dysfunction, and other adverse pathophysiological health effects [18].

Given the complex interactions and influence of genetic, epigenetic, and environmental factors (e.g., diet, pollution, medications, physical activity, infections, and stress) on the gut–brain axis, therapeutic strategies that target the gut microbiome and intestinal environment to promote gut and mental health while limiting adverse physiological effects and associated disease risk are needed. Due to their direct interactions with the intestinal environment, dietary strategies are particularly promising in this regard.

Dietary interventions, including dietary supplements, are increasingly utilized by practitioners and the public to support gut and mental health. Certain phytochemicals such as polyphenols and omega fatty acids exert prebiotic-like effects and can increase the growth of beneficial microbes in the gut while also reducing the growth of detrimental microbes [19]. Additionally, phytochemicals such as polyphenols and carotenoids have been demonstrated to have therapeutic potential in targeting the gut–brain axis, as well as outcomes including stress and anxiety [20]. For instance, a study examining the effects of supplementation with carotenoids, including lutein, in young adults found that these phytochemicals reduced psychological stress and serum cortisol levels [21]. In this context, marine-derived ingredients from microalgae are garnering increasing interest as sources of nutrients and phytochemicals with potential health benefits [22,23,24,25]. Microalgae contain numerous nutrients (e.g., vitamins, minerals, polyunsaturated fatty acids) and phytochemicals with bioactive properties including pigmented compounds (e.g., carotenoids, chlorophyll), polyphenols, and sterols [26]. Specifically, recent research suggests that microalgae extracts from *Phaeodactylum tricornutum*, which contain the carotenoid fucoxanthin, can enhance cognitive function, physical performance, bone health, and lipid profiles in various human studies [27,28,29]. However, no human studies have examined the effects of microalgae extract supplementation on gut health and its connection to mental health (i.e., the gut–brain axis), despite promising preclinical findings from various microalgae strains [30,31]. Notably, studies in different mouse models have demonstrated that microalgae extracts from *Spirulina platensis*, *Dunaliella salina*, *Chlorella vulgaris*, and *Haematococcus pluvialis* may alleviate symptoms of constipation and rectal bleeding by reducing colon tissue mucosal damage, inflammation, and oxidative stress, while modulating gut microbiota [22,32]. Extracts from *T. obliquus*, one of the most widely produced species in Europe [33], have also shown potential gut health benefits, although the majority of the current literature focuses on its production processes. A recent study in healthy dogs demonstrated that supplementation with *T. obliquus* extract in kibble increased specific taxa abundances [34]. Compared to other microalgae strain such as *Spirulina platensis* or *Chlorella vulgaris* which are rich in proteins, *T. obliquus* strain Mi175.B1.a is recognized to contain high levels of lipids (including PUFAs and MUFAs), carbohydrates, and carotenoids, like lutein [35]. These bioactives may directly influence the composition of the gut microbiome, enhance SCFA production, and modulate intestinal permeability and gut barrier function [36,37,38]. These data suggest that dietary supplementation with microalgae extract has promise for promoting gut health and other aspects of health linked to the gut, such as mental health, and underscore the need for further investigation in human studies.

The purpose of the present study was to investigate whether four weeks of supplementation with *Tetradesmus obliquus* strain Mi175.B1.a microalgae extract (TOME; 250 mg/day) influences gut health and subjective and physiological indicators related to stress and anxiety in healthy adults experiencing mild to moderate GI complaints. By improving overall gut properties, microalgae supplementation could regulate biomarkers involved in stress/anxiety, inflammation, and other indicators of health. Therefore, we hypothesized that TOME supplementation would improve self-perceived GI symptoms, and increase stool consistency, gut integrity, and microbial diversity. We also hypothesized that TOME supplementation would improve blood pressure-related responses to sympathetic nervous system activation induced by the cold pressor test (CPT).

## 2. Methods

### 2.1. Research Design

The study was conducted as a randomized, double-blind, placebo-controlled, parallel-arm intervention trial (Figure 1) at Colorado State University (CSU)’s Food and Nutrition Clinical Research Laboratory (FNCRL) within the Department of Food Science and Human Nutrition. Briefly, healthy adults with mild to moderate GI distress were recruited from Fort Collins, Colorado and surrounding areas through referrals from local healthcare providers, flyers, emails, and word of mouth. Initial eligibility was assessed via a phone screening questionnaire and confirmed onsite at the FNCRL. Healthy, free-living adults aged 18–55 years with a BMI between 18.0–29.9 kg/m^2^ who experienced mild to moderate GI distress, confirmed by one positive response to the Rome IV criteria, were enrolled in the study. Exclusion criteria included the following: (A) individuals <18 or >55 years of age; (B) BMI outside the 18.0–29.9 kg/m^2^ range; (C) pregnancy or breastfeeding; (D) current smoker or use of tobacco products; (E) diagnosis of intestinal diseases such as Celiac disease, Crohn’s disease, Ulcerative Colitis, or GI cancers; (F) regular use of statins, metformin, steroids, NSAIDs, or MAO inhibitors; (G) a clinical diagnosis of mental health disorders (e.g., depression, bipolar disorder, Alzheimer’s disease, etc.); (H) a known allergy to any ingredients in the microalgae extract or maltodextrin; and (I) a history of alcohol or substance abuse within the past 12 months. Participants were asked to maintain their regular eating and exercise habits throughout the study duration.

Eligible participants were enrolled and given sequential study codes, then were randomized to consume two powder-encased capsules per day of either a maltodextrin placebo (Placebo) or 250 mg of a microalgae *Tetradesmus obliquus* strain Mi175.B1.a (TOME) via randomization codes generated in Matlab version R2021b (MathWorks, Natick, MA, USA) and printed on capsule packages. Study participants, principal investigators, and clinical personnel (FNCRL, CSU) were blinded to the treatment groups until all primary data analyses were complete.

Participants attended clinic visits for eligibility screening and at baseline, 2-week, and 4-week time points. Participants arrived at the clinic after fasting for at least eight hours, and refraining from strenuous exercise, alcohol consumption, and use of medications or dietary supplements for at least 24 h. During screening, medical history, health status, and demographic information were collected, followed by anthropometric measurements (height, weight, and waist and hip circumferences). Participants meeting all eligibility criteria were asked to complete a 24 h dietary recall on three separate days (two weekdays and one weekend day) prior to the baseline and final study visits, using the National Cancer Institute’s Automated Self-Administered 24 h dietary assessment tool (ASA24; https://asa24.nci.nih.gov). Participants were instructed to record their bowel movements using the Bristol Stool Scale (BSS) for at least 7 days prior to their baseline clinic visit to capture their normal stool habits. At the baseline, 2-week (midpoint), and 4-week (final) visits, in addition to anthropometric and blood pressure measurements, participants were asked about their physical activity levels over the previous 7 days using a validated questionnaire as described [39]. They also completed several questionnaires, including the Gastrointestinal Symptom Rating Scale (GSRS), the Generalized Anxiety Disorder 7-item scale (GAD-7), the Positive and Negative Affect Schedule (PANAS), and the Perceived Stress Scale (PSS). Blood and saliva samples were collected following standard procedures to analyze biomarkers related to GI health, stress, inflammation, and metabolic/lipid panels. Participants were also instructed to collect a stool sample at home the day before each study visit using a provided kit. Stool samples were used to assess fecal markers of intestinal permeability and function. At the end of the baseline visit, participants received their four-week treatment supply and a daily treatment log. At the final visit, participants returned unused treatment capsules to assess compliance. Primary outcomes included safety and tolerability of the product and indicators of GI health, including changes in GI symptoms and bowel movements, microbiota composition, gut function, and permeability biomarkers. Secondary outcomes included assessing impacts on stress and anxiety through mental health assessments and acute stress biomarkers, blood pressure and pulse wave analysis assessment at baseline and during a cold pressor test (CPT), and changes in lipid profiles. Finally, responses to inflammation were determined by measuring inflammatory biomarkers in LPS-stimulated PBMCs. The trial is registered at ClinicalTrials.gov (NCT06425094) and was approved by the CSU Institutional Review Board (#5162), in accordance with the Declaration of Helsinki.

### 2.2. Intervention and Compliance

Participants were randomly assigned to take supplements containing either a Placebo or TOME daily for four weeks. The experimental supplement consisted of two capsules daily, each containing 125 mg of powdered microalgae extract (equivalent to 50 mg of liquid form per capsule), standardized to 0.3% lutein (Zengut™, Microphyt, Baillargues, France). Microalgae supplement characterization was carried out by the Eurofins and ITERG companies and is described in Table 1. The Placebo capsules were identical in appearance, containing 125 mg of maltodextrin, and were formulated to taste the same as the experimental supplement. Participants ingested two capsules daily. The product manufacturers provided a certificate of analysis to verify the dosage and confirm the absence of contaminants. Supplementation began on the first study day following baseline testing, and participants were instructed to take the capsules daily at lunchtime with eight ounces of water. Both the Placebo and TOME supplements were distributed in blister packs and stored at 4 °C. To assess treatment compliance, participants completed treatment diaries and returned any unused capsules at their final study visit.

### 2.3. Anthropometrics

Height (cm) was measured using a scale-mounted stadiometer to the nearest 0.5 cm, and weight (kg) was measured with a digital scale (Health-O-Meter Professional Scale, Sunbeam Products, Inc., McCook, IL, USA). Waist and hip circumferences were recorded using a Gulick measuring tape with a tension handle (Creative Health Products, Inc., Ann Arbor, MI, USA) in accordance with standard procedures [41].

### 2.4. GI Disorder and Bowel Movement Assessments

Participants completed the Gastrointestinal Symptom Rating Scale (GSRS) questionnaire at baseline, week 2 and week 4. Briefly, the GSRS is a disease-specific questionnaire consisting of 15 items grouped into five symptom clusters: reflux, abdominal pain, indigestion, diarrhea, and constipation. It uses a seven-point Likert scale, where 1 indicates no troublesome symptoms, and 7 indicates extremely troublesome symptoms. Participants were instructed to log all bowel movements throughout the study and classify them using the Bristol Stool Chart. A copy of the chart was provided as a reference to assist with identifying stool types. Soft stools were classified as normal, while hard or diarrheal stools were considered abnormal.

### 2.5. Mental Health Assessments

Participants completed three different questionnaires to evaluate mood, anxiety, and stress at baseline and weeks 2 and 4. The Positive and Negative Affect Schedule (PANAS) was used to assess mood [42]. This self-reported scale evaluates two broad dimensions of affect (mood) on a 5-point Likert scale, ranging from “not at all” to “very much” for each item. Positive Affect reflects the degree to which a person feels enthusiastic, energetic, or alert, with low scores indicating sluggishness. Negative Affect captures subjective distress, including emotions such as anger, fear, disgust, nervousness, and contempt, with low scores indicating calmness and serenity. Separate scores for Positive and Negative Affect were calculated by summing the ratings for words associated with each dimension. The Generalized Anxiety Disorder 7-item (GAD-7) questionnaire screens for anxiety symptoms over the past four weeks, with total scores ranging from 0 to 21, based on individual item scores of 0 to 3 [43]. Scores of 0–4 indicate minimal anxiety, 5–9 reflect mild anxiety, 10–14 indicate moderate anxiety, and 15 or above suggest severe anxiety [43]. Cohen’s Perceived Stress Scale (PSS) was used to measure participants’ perceptions of stress. The PSS assesses the degree to which individuals feel their lives are unpredictable, uncontrollable, or overwhelming. Participants were asked how often they had experienced specific emotions or thoughts during the past month. The scale has demonstrated strong test-retest reliability, with values exceeding 0.70 [44].

### 2.6. Hemodynamics and Pulse Wave Analysis at Rest and During CPT

Brachial systolic blood pressure and diastolic blood pressure were measured in triplicate (and averaged) on the upper left arm in the supine position after 10 min of rest, using an automated blood pressure monitor (Omron HEM-907XL Professional Blood Pressure Monitor, Hoffman Estates, IL, USA). Pulse wave analysis was performed in triplicate (and averaged) following brachial blood pressure measurement using the SphygmoCor XCEL (AtCor Medical Inc., Naperville, IL, USA) as described [45]. Outcomes for pulse wave analysis included aortic systolic and diastolic pressure, mean arterial pressure (MAP), pulse pressure, aortic pressure (AP), heart rate, augmentation index (AIx), and AIx normalized to a heart rate of 75 bpm (AIx@75). For the CPT, participants submerged their right hand just above the wrist into an ice/water slurry bath (1–4 °C confirmed with a digital thermometer and water kept circulating with a pump) for 2 min. Brachial blood pressure measurements were performed at 1 and 2 min time points during the CPT, while pulse wave analysis measurements were performed at the 2 min time point during the CPT. After the 2 min period, the participant’s hand was removed from the bath and wrapped in a towel, and brachial blood pressure was measured every 1 min for a total of 3 min.

### 2.7. Biological Specimen Collection

Blood, saliva, and stool samples were collected from participants at each study visit. Venous blood samples were collected in lithium heparin and ethylenediaminetetraacetic acid (EDTA) tubes. Plasma was collected by centrifugation of the EDTA tubes and stored at −80 °C. Two hundred microliters of lithium heparin whole blood was analyzed for lipid and metabolic biomarkers using the Piccolo Comprehensive Metabolic Panel and Lipid Panel on a Piccolo Xpress blood chemistry analyzer (Abaxis Inc., Union City, CA, USA). PBMCs were isolated and stored in liquid nitrogen as described [46]. Saliva samples were collected as previously described [45,47] and stored at −80 °C until use. Participants provided a stool sample, which was collected at home either the day prior to or the morning of their clinic visit. Participants were instructed to keep stool samples frozen until delivery. Once delivered, samples were stored at −80 °C until analyzed for microbiome and gut health indicators.

### 2.8. Microbiome 16s rRNA Gene Sequencing and Processing

The FastDNA^TM^ SPIN Kit for feces (MP Biomedicals, Solon, OH, USA) was used to extract microbial DNA from participant fecal samples, according to the manufacturer’s instructions. Amplicon libraries were generated for the 16s rRNA hypervariable V4 region using 515F-806R primer set according to protocols published for the Earth Microbiome Project (16S Illumina Amplicon Protocol: earthmicrobiome) [48]. DNA extraction controls, no template PCR controls, and the Zymo mock gut microbial community were included on the sequencing plate as quality controls. Amplicon sequencing was performed on an Illumina MiSeq (Illumina Inc., San Diego, CA, USA) using 2 × 250 bp paired-end reads by the Next Generation Sequencing Core Facility at CSU (Fort Collins, CO, USA). The resulting 16S rRNA amplicon dataset was processed using QIIME2 (v2023.5) [49]. The DADA2 pipeline within QIIME2 was used to quality check and trim reads with a quality score of greater than 30, followed by denoising and clustering amplicon sequence variants (ASVs). Clustered ASVs were taxonomically classified using the SILVA database v138. The trimmed and denoised data was exported from QIIME2 and converted into a comma-delimited file. The data were then filtered and normalized in MicrobiomeAnalyst [50]. To remove features that may be a result of sequencing error or low-level contamination, a low count filter removed reads with less than four counts and reads that were present in less than 10% of the samples; standard deviation was used as a low variance filter. One sample with <10,000 reads was removed from further analysis. Data were then normalized via total sum scaling. The resulting filtered and normalized data were used in downstream analyses.

### 2.9. Short-Chain Fatty Acid (SCFA) Extraction and Analysis

SCFAs were extracted from fecal samples and quantified by Gas Chromatograph with Flame Ionization Detection (GC-FID) as previously described [51]. Briefly, one gram of feces was sonicated in acidified, HPLC-grade water (pH brought to 2.5 with HCl) and allowed to sit at RT for 10 min. Samples were then centrifuged, and supernatants removed and frozen overnight at −80 °C. The supernatants were then thawed, centrifuged again to sediment any remaining particulate matter, and filtered through 0.45-micron filters. Supernatants were analyzed on a GC-FID (Agilent 6890 Plus GC Series, Agilent 7683 Injector Series, GC Column: TG-WAXMS A 30 m × 0.25 mm × 0.25 μm). Quantities of specific SCFAs (acetate, propionate, and butyrate) were determined through generation of standard curves of purified commercial chemicals.

### 2.10. Blood, Saliva, and Stool ELISA Assays

Biomarkers of stress, sympathetic activation, and intestinal function were measured in stool, saliva, and/or plasma using ELISA assays according to manufacturer’s instructions. Stress markers included the following: plasma and salivary cortisol (MP Biomedicals, Solon, OH, USA), plasma and salivary a-amylase (Novus Biologicals, Centennial, CO, USA), plasma and salivary chromagranin A (Novus Biologicals, Centennial, CO, USA), plasma blood-derived neurotrophic factor (BDNF; BioLegend, San Diego, CA, USA), and plasma adrenocorticotropic hormone (ATCH; Eagle Biosciences, Amherst, NH, USA). Markers of intestinal barrier function and inflammation included stool measurement of secretory IgA (sIgA; Immuchrom, Heppenhein, Germany), zonulin (AFG Bioscience, Northbrook, IL, USA), and a1-antitrypsin (AAT; ALPCO, Salem, NH, USA). Standard curves were used to quantify the concentration of each inflammatory marker.

### 2.11. PBMC Culturing and Stimulation

PBMC processing was performed in a laminar flow hood. A complete culture medium containing 1× RPMI-1640 (Corning; Corning, NY, USA), 10% FBS (Atlas Biologics; Fort Collins, CO, USA), and 1% penicillin/streptomycin [100 U/mL penicillin and 100 μg/mL streptomycin] (HyClone; Tianjin, China) was warmed to 37 °C in a water bath, which was also used to thaw the frozen PBMCs. Samples were transferred to a 15 mL conical tube and warm medium was added at a rate of 1 mL/5 s to a final volume of 10 mL. Cells were centrifuged for 8 min at 300× *g* at 25 °C, decanted, and washed twice. Cells were then plated for an overnight recovery period at about 1–2 × 10^6^ cells/mL. After the 24 h rest, cells were counted and seeded in triplicate at a desired concentration of ~5 × 10^5^ cells/well in 96-well plates. The wells were treated with 1 μg/mL (final concentration) of *E. coli* LPS (Sigma-Aldrich; St. Louis, MO, USA) and incubated at 37 °C and 5% CO_2_ for 24 h. Post incubation, supernatants were collected and analyzed for inflammatory markers using the QPlex Human T Cell Cytokine HS (9-Plex) kit (Quansys Bio, Logan, UT, USA) according to manufacturer’s instructions.

### 2.12. Lipid and Comprehensive Metabolic Panels

Blood samples (100 μL) were collected during each study visit from the antecubital vein in a lithium heparin tube and immediately analyzed using a Piccolo Metlyte Plus CRP Reagent Disc, which included glucose, blood urea nitrogen (BUN), creatinine (CRE), Sodium, Potassium, Chloride, Carbon dioxide, Calcium, Total proteins, Albumin, Bilirubin, Alkaline phosphatase, Alanine transaminase, and Aspartate aminotransferase on the Piccolo Xpress Chemistry Blood Analyzer (Abaxis, Union City, CA, USA). Lipid panels (total cholesterol (TC), high-density lipoprotein cholesterol (HDL), triglycerides (TRIGs), total cholesterol/HDL ratio (TC/HDL), low-density lipoprotein cholesterol (LDL)) were assayed within one hour of blood collection using a Piccolo Xpress Chemistry Blood Analyzer (Abaxis, Union City, CA, USA).

### 2.13. Statistical Analyses

#### 2.13.1. Analysis of Biomarkers and Participant Outcomes

As an exploratory study, we used other studies looking at the impact of dietary supplements on self-reported gut health parameters to estimate the sample size needed [31,52,53]. These studies showed that sample sizes of 20–30 participants per group were sufficient to detect significant differences among the gut health parameters that were evaluated. Thus, we targeted enrollment of 25 people per group to obtain sufficient data for conducting sample size calculations that could be used in powering future studies to the various outcomes that we measured. All statistical analyses related to inflammatory biomarkers (fecal and serum) and participant-reported outcomes were performed using GraphPad Prism 10 Version 10.2.2. Data were assessed for normality using QQ plots and Shapiro–Wilk statistical tests. All inflammatory biomarker outcomes and participant-reported outcomes were analyzed using linear mixed effects models that included *group*, *time*, and *time × group* as fixed effects and *subject* as a random effect. Any biomarkers and participant-reported outcomes showing significant (*p* < 0.05) relationships with fixed covariates were brought forward into additional analyses (e.g., multiple comparisons test and change from baseline). *Post hoc* multiple comparisons were conducted using Tukey’s or Sidak’s tests; accordingly, all reported *p*-values for the multiple comparisons are adjusted *p*-values. All mixed effects models were run as *per protocol* (PP; n = 53). Two-tailed *t*-tests or non-parametric Wilcoxon Rank Sum tests were used to determine if there were any differences in change from baseline values between the microalgae extract and comparator treatments. For all statistical tests, *p*-values < 0.05 were considered statistically significant, with *p*-values < 0.1 being reported as non-significant trends. Effect sizes for primary outcomes (self-reported GI symptom severity scores and mental health assessments) were determined using partial ETA squared (η_p_^2^), which was calculated as the SS_effect_/(SS_effect_ + SS_error_). Calculated values range from 0–1 with values between 0.01–0.05 considered a small effect, 0.06–0.13 a medium effect, and >0.14 a large effect.

#### 2.13.2. Microbiome Analyses

Alpha diversity measures (Shannon Index, Observed Richness, and Chao1) for each time point and treatment were calculated using MicrobiomeAnalyst [50] and statistically significant differences were determined using Mann–Whitney *post hoc* tests with Benjamini–Hochberg adjustment for false discovery (FDR). A Principal Coordinate Analysis (PCoA) based on Bray–Curtis dissimilarity metrics was used to observe differences in bacterial communities between the microalgae extract and comparator groups. Differences in Bray–Curtis distances were statistically analyzed using a permutational analysis of variance (PERMANOVA) as well as pairwise PERMANOVAs. Differential abundance in specific taxa between groups or over time was determined by looking for consensus taxa identified by Analysis of Microbiomes with Bias Correction (ANOCOM-BC) [54], Microbiome Multivariable Association with Linear Models (MaAsLin2) [55], and Linear Discriminate Analysis Effect Size (LEfSe) [56].

## 3. Results

### 3.1. Participant Characteristics

A total of 145 individuals responded to study advertisements and were assessed for eligibility. Of these, 68 individuals passed the online pre-screening and were invited for the clinical screening visit (CONSORT Figure 2).

At this visit, individuals were familiarized with the study and provided written informed consent. A total of 56 individuals met all criteria and were enrolled into the intervention. A total of 28 participants were randomized into the Placebo group and 28 into the TOME group. A total of 53 participants completed the study and were included in the statistical analyses (*Per Protocol*). Participants who did not complete the study dropped out because of inability to attend scheduled clinic visits (n = 2), and a change in medication (n = 1). Participant demographics are included in Table 2.

#### Compliance, Safety, Tolerability, and Diet

Compliance with the intervention protocol was determined by calculating the percentage of consumed capsules. Average compliance for the Placebo group was 97% (range = 82–100%) and average compliance for the TOME group was 94% (range = 75–100%). To accurately assess the impact of the intervention, participants were instructed to adhere to their habitual diet and physical activity regime. No significant differences in total caloric, macronutrient, or micronutrient intake were noted between or within groups at any time point (Appendix A).

Study-related adverse events were reported to be mild/moderate and included one report of constipation, one report of diarrhea, one report of bloating and dizziness, and three reports of increased anxiety in participants on the Placebo. Treatment-related adverse events included six reports of increased flatulence, four reports of increased bloating, two reports of borborygmi, and one report of increased abdominal pain. No participants withdrew from the study due to adverse events.

Metabolic panels for liver function and blood chemistry remained within standard reference ranges throughout the study, although there was a significant main effect of *time* observed for glucose (*p* = 0.03), sodium (*p* = 0.03), AST (*p* = 0.002), and albumin (*p* = 0.002) (Appendix A). *Post hoc* analyses revealed that AST was significantly reduced between baseline and week 4 in the TOME group (*p* = 0.03) and albumin was significantly reduced between baseline and week 4 in the Placebo group (*p* = 0.01). Overall, these findings suggest that the microalgae extract was safe and well-tolerated in this population.

### 3.2. Gut Function

#### 3.2.1. GI Symptom Severity and Bowel Movement Assessments

The scores for the GSRS self-assessment tool indicated a reduction in GI symptom severity in participants consuming TOME capsules for 4 weeks (Appendix A). There was a significant main effect for *time* (*p* < 0.001; η_p_^2^ = 0.20, large effect) and a significant *time × group* interaction effect (*p* = 0.037; η_p_^2^ = 0.06, medium effect). The *post hoc* Tukey’s test adjusted for multiple comparisons showed that there were improvements in score with both Placebo (*p* = 0.02) and TOME (*p* = 0.01) at 2 weeks; however, these improvements only persisted in the TOME group after 4 weeks (Figure 3A; *p* = 0.002). The GSRS score can be subdivided into five sections relating to different symptomologies: abdominal pain, constipation, diarrhea, indigestion, and reflux. There were no significant changes in reflux scores at 4 weeks for either group, but there were significant improvements in each of the remaining four symptomologies. For constipation scores, there were significant effects for both *time* (*p* = 0.048, η_p_^2^ = 0.06, medium effect) and *group* (*p* = 0.047; η_p_^2^ = 0.16, large effect) and a trend towards a significant interaction effect (*p* = 0.08; η_p_^2^ < 0.01, no effect). These were driven by significant baseline differences between the groups (*p* = 0.003) as well as reduced constipation-related symptoms in the TOME group at 2 weeks (*p* = 0.04) and 4 weeks (Figure 3C; *p* = 0.001). For diarrhea-related scores, there was a significant effect of *time* (*p* = 0.003, η_p_^2^ = 0.11, large effect) as well as significant improvement between baseline and 4 weeks in the TOME group (Figure 3D; *p* = 0.038). Finally, there was a significant effect of *time* on indigestion (*p* = 0.078, η_p_^2^ = 0.16, large effect) and a trend for a *time × group* interaction (*p* = 0.078, η_p_^2^ = 0.05, small effect). These effects were driven by improvements in the TOME group between baseline and the 2-week (*p* = 0.04) and 4-week (Figure 3E; *p* = 0.001) timepoints (Appendix A). Finally, the analysis of mean changes from baseline showed that total score (*p* = 0.017), indigestion score (*p* = 0.031), and constipation score (*p* = 0.048) were significantly decreased in TOME group compared to Placebo after 4 weeks of supplementation, while no significant difference was seen in scores for abdominal pain, diarrhea, and reflux (Appendix A).

Although there were self-reported differences in both diarrhea and constipation scores for the GSRS, we saw no significant changes in reported bowel habits using the Bristol Stool Scale. Specifically, we saw no changes in either the mean score, ratio of total to abnormal stools (scored as 1, 2, 6, or 7), or in ratios of bowel movements reported as either constipation or diarrhea (Appendix A).

#### 3.2.2. Gut Microbiota Assessment

The average number of reads per sample was 36,549 and the minimum sample size was 12,500. Microbial richness, determined by observed ASVs and CHAO estimations, was identical, and did not differ significantly between groups or over time (Figure 4A). However, there were differences in Shannon’s diversity index (*p* = 0.009; Kruskal–Wallis statistic = 15.43). Pairwise comparisons revealed that the majority of these differences occurred between the TOME and Placebo groups (Appendix A; Figure 4B), which is unsurprising given that microbiomes differ more between people than they do within people over time. In the Placebo group, there were no longitudinal changes in Shannon’s diversity over four weeks. However, in the TOME group, there was an increase in Shannon’s diversity between baseline and week 4, although it was not significant after adjusting for False Discovery Rate (Figure 4B; *p* = 0.046; FDR = 0.115). Beta-diversity was visualized using Bray–Curtis distances and showed no clustering or significant separation by either time or treatment group (Figure 4C). There were also no differentially abundant taxa identified by ANCOM-BC, MaAsLin, or LeFSE (LDA > 2.0).

#### 3.2.3. Markers of Gut Barrier Integrity and Microbial Activity

Gut barrier integrity can be measured indirectly by determining the presence of various proteins in blood or stool. Fecal zonulin, a protein responsible for reversibly modulating tight junctions, showed a significant effect for *group* (*p* = 0.04), but no significant differences were detected by *post hoc* pairwise comparison (Figure 5A). The change from baseline in fecal zonulin was not significant between groups (*p* = 0.39), so the observed group effect was likely driven by the slightly lower average zonulin levels observed at both timepoints in the TOME group. For fecal alpha-1 antitrypsin, a protease inhibitor expressed in the liver, there was a significant *time × group* interaction for AAT (*p* = 0.005) and *post hoc* comparisons showed that this was driven by a significant decrease in AAT from baseline to week 4 in the Placebo group (Figure 5B; *p* = 0.02). As shown in Figure 5C, we observed a high level of variability in sIgA levels among participants, and did not see any significant effects of the intervention.

Using gas chromatography, we measured fecal SCFAs at each timepoint during the intervention as a marker of gut microbial metabolism. We did not detect significant changes in SCFA levels either between (*group*) or within *(time*) the intervention groups (Figure 5D–F), with the exception of a significant *time × group* interaction for butyrate (*p* = 0.024). Using Sidak’s *post hoc* analysis, there was a significantly different concentration of butyrate between the treatment groups at week 4 (Figure 5F; *p* = 0.039).

### 3.3. Effects on Mental Health Parameters

#### 3.3.1. Self-Reported Assessments of Affect, Anxiety, and Stress

Several questionnaires were used to assess whether the intervention impacted anxiety (GAD-7), stress (PSS), and mood (PANAS). Both groups reported low levels of anxiety at baseline according to GAD-7 scores (TOME $\bar{x}$ = 5.3; Placebo $\bar{x}$ = 4.7), and there were no significant differences between groups or over time (Figure 6A). At baseline, participants’ average PSS scores fell into the mild/moderate perceived stress range (TOME $\bar{x}$ = 14.46; Placebo $\bar{x}$ = 13.92). The mixed effects model showed a significant effect of *time* (Figure 6B, *p* < 0.001) and *post hoc* analysis showed that there was a significant reduction in perceived stress in both the TOME (*p* = 0.028) and Placebo groups (*p* = 0.014) at week 4 compared to baseline. PANAS results for Positive Affect Scores showed a significant effect for *time* (*p* = 0.010) with *post hoc* comparisons showing a significant decrease in Positive Affect in the TOME group between weeks 2 and 4 (Figure 6C; *p* = 0.003). There were no other statistically significant comparisons; however, at week 2, there was a trend towards significance in the difference between the TOME group and Placebo (*p* = 0.059). For the Negative Affect Scores, there was a significant effect of *time* (*p* = 0.006) as well as a *group × time* interaction effect (*p* = 0.021; Figure 6D). Specifically, there was a significant decrease in Negative Affect Scores in the TOME group between weeks 2 and 4 (*p* = 0.009) and between baseline and week 4 (*p* < 0.001). To further explore the differences that we observed in Positive and Negative Affect scores, we compared changes at week 2 and week 4 from baseline and between groups. The change from baseline between weeks 2 and weeks 4 in the TOME group for Positive Affect (Figure 6E; *p* = 0.001) and Negative Affect (Figure 6F; *p* = 0.005) were statistically significant. In addition, the change in Negative Affect at the end of the study was significantly different compared to the Placebo (*p* = 0.033).

#### 3.3.2. Blood and Saliva Markers of Stress

Several biological markers of stress and anxiety were measured in saliva and/or plasma and presented in Appendix A. Cortisol, a stress hormone produced by the adrenal glands, and alpha(α)-amylase and chromagranin A, markers of the sympathetic-adrenal-medullary activation of sympathetic nervous system, were measured in both the saliva and plasma. There were no differences between groups or over time for cortisol in either the saliva or the plasma. There were also no significant differences in salivary alpha-amylase, but there was a significant effect of *time* (*p* = 0.04) for plasma α-amylase. Pairwise comparisons showed that there was a trending decrease in α-amylase in the Placebo group (Figure 7A; *p* = 0.09). In addition, there was a trending effect for *time* in plasma chromagranin A levels (*p* = 0.06). *Post hoc* analysis showed that chromagranin A was significantly decreased after TOME intervention at week 4 compared to baseline (Figure 7B; *p* = 0.03), as well as a trending decrease compared to the Placebo group (*p* = 0.08). Salivary chromagranin A did not differ significantly over time or among groups. We also measured BDNF and ACTH in plasma. BDNF serves to protect neurons and tends to decrease with chronic stress, while ACTH is produced by the pituitary glands and signals the body to produce cortisol. Neither of these markers were significantly altered by either TOME or Placebo over the course of the study.

### 3.4. Cardiovascular-Related Outcomes

Blood pressure and other parameters assessed through pulse wave analysis are shown in Appendix A. At baseline and before the CPT (i.e., resting), there were no statistically significant differences between groups. However, the Placebo group had a significantly (*p* = 0.021) lower augmentation index (AIx) response at the 2 min timepoint during the CPT at baseline and the week-4 timepoint compared to the TOME group, suggesting baseline differences in cardiovascular function. Resting aortic systolic blood pressure significantly increased (*p* = 0.028) in the Placebo group but not in the TOME group from baseline to 4 weeks, and there were no differences between groups. Brachial diastolic blood pressure responses after 2 min of the CPT were significantly higher at week 4 than baseline in the TOME group but not Placebo, though there were no differences between groups. At week 4, the TOME group had higher AIx@75 at the 2 min CPT than the Placebo group (*p* = 0.034). No other significant differences were observed within or between groups.

Blood lipids remained within acceptable ranges throughout the study in both intervention groups (Appendix A). The only parameter that significantly varied during the study was triglycerides, which had a significant effect for *time* (*p* = 0.018) and trending significance for *time × group* interaction (*p* = 0.051). Triglycerides were significantly increased in the TOME group at week 4 compared to baseline (*p* = 0.026) and week 2 (*p* = 0.004). However, the average triglyceride values at week 4 did not differ significantly between the TOME and Placebo groups (*p* = 0.166), nor were they above the clinically acceptable range of 150 mg/dL.

### 3.5. Markers of Systemic Inflammation and Immune Function

To assess inflammation and immune responses, we measured nine cytokines and inflammatory markers in supernatants from LPS-stimulated PBMCs (Appendix A). Markers included TNF and IFN-gamma as well as IL-2, IL-4, IL- 5, IL-10, IL-13, IL-17, and IL-21. There were no significant differences either within or between intervention groups, except for IL-13, which had a significant *time × treatment* interaction (*p* = 0.042). *Post hoc* analysis showed no significant pairwise differences, but there was a trend for higher IL-13 at baseline in the Placebo group compared to the TOME group (*p* = 0.07).

## 4. Discussion

Marine-derived natural ingredients, such as microalgae, have emerged as rich sources of bioactive compounds that could serve as sustainable sources of nutrition for humans and animals [57]. Among these, extracts from the microalgae *Tetradesmus obliquus* are rich in phytochemicals such as lutein and polyphenols and contain omega-3 fatty acids and other lipids that could potentially modulate gut microbiota structure and influence intestinal function [24,57]. This study assessed the impact of supplementation with *Tetradesmus obliquus* strain Mi175.B1.a (TOME; 250 mg/day) for four weeks on subjective and physiologic indicators of gut and mental health in healthy adults experiencing mild to moderate GI complaints. To the best of our knowledge, this is the first investigation to establish the safety and tolerability of TOME supplementation in humans while demonstrating its impacts on GI and mental health. These findings indicate that TOME supplementation improves GI symptoms, indicated by a reduction in GSRS global scores, with notable improvements in the constipation and indigestion sub-categories, potentially through effects on the gut microbiota. Furthermore, TOME exhibited positive effects on mental health. Here, we contextualize these findings and discuss study limitations and future directions for this research.

### 4.1. Gut Health, Microbiota, and Intestinal Function

Microalgae can have positive effects on gut health due to their high fiber, phytochemical, and omega-3 content, which can reduce intestinal inflammation and serve as prebiotics to modulate gut microbiota and improve gut barrier integrity [58,59,60]. We observed a significant reduction of around 20% in self-reported GI symptoms and in particular the constipation and indigestion scores. To date, there is only one other published study that explored the effects of a microalgae preparation on gut health in humans. In that study, *Chlamydomonas reinhardtii* was reported to improve GI symptoms, consistent with the results of the current study. In contrast, they also reported improvements in stool form and bowel habits which were not observed in the current study [52]. Although the population and duration of microalgae consumption was similar between these studies, the *C. reinhardtii* study used much higher doses (1 or 3 g) of microalgae and included whole cell preparations that would retain more insoluble fiber content than extracts used in the current study. In addition, they used recall data to capture bowel habits rather than more accurate daily stool diaries and the reported benefits were only observed in a subgroup with higher baseline levels of GI symptoms. In the current study, participants reported that about 20% of their bowel movements were abnormal (including both diarrhea and constipation), and there was a high level of variability in individual responses. Future studies could be powered to separately explore the effects of TOME on diarrhea and/or constipation to better determine whether there are benefits for bowel habits. Regardless, the improved GSRS sub-scores for constipation and diarrhea indicate that participants perceived benefits in bowel habits.

In the current study, we explored the impact of TOME supplementation on gut microbiota composition. Although no differences in β-diversity or specific taxa changes were observed, we noted an increase in Shannon’s diversity within the TOME group but not the Placebo, although this increase was not significant after Benjamini–Hochberg adjustment (FDR = 0.115). Interestingly, the baseline diversity in the TOME group was significantly lower than the Placebo group, but there were no significant differences between the two groups at 4 weeks. In the Fields et al. study utilizing *C. reinhardtii*, they did not report any changes in microbial diversity with microalgae supplementation, but did observe significantly lower diversity in the subgroup with high GI distress compared with the subgroup with lower GI distress [52]. Although we did not stratify the analysis by GI distress level, we did observe much higher constipation scores in the TOME group at baseline compared to the Placebo, which may be related to the reduced levels of baseline microbiota diversity in this group. In general, higher gut microbial diversity is associated with more resilience to perturbation and has also been associated with benefits to mental health [61]. Therefore, future studies should be designed to capture the effects of TOME on microbiota composition in subgroups with varying types and levels of GI complaint. Finally, 16s rRNA sequencing data is limited to taxonomic classification of the microbiota and does not capture changes in microbial gene expression, which may result in altered profiles of bioactive metabolites or exert influences on mucosal/systemic immunity that can confer benefits to the host. The addition of metagenomic or meta-transcriptomic approaches in future studies would better elucidate microbiota-modulating effects.

In the absence of microbial gene expression data, we used markers of intestinal barrier function (zonulin and alpha-1-antitrypsin), mucosal immunity (sIgA), and microbial metabolism (SCFA) as proxy measures for microbiome function. There were no significant changes in these markers that could directly be attributed to TOME treatment; however, there was a significant main effect of *group* (*p* = 0.04) for fecal zonulin. Zonulin is a protein responsible for reversibly modulating tight junction activity by disassembling the tight junctions between epithelial cells and increasing barrier permeability [62]. Average zonulin decreased with TOME and increased in the Placebo group, so it is possible that there were beneficial effects of TOME gut barrier function that the study was not powered to detect. In contrast, there was a significant decrease in alpha-1-antitrypsin (AAT) in the Placebo group which was not observed with TOME supplementation. AAT is a protease inhibitor that is expressed in the liver and inhibits neutrophil elastase [63]. It is transported via circulation to target tissues and can leak into the intestine when barrier function is compromised [64], suggesting improved barrier function in the Placebo group. In humans, a more definitive measure of intestinal barrier function is the lactulose: mannitol (LM) test, which is a quantitative assay for directly measuring intestinal permeability. While this test significantly increases participant burden, it would also provide a much more definitive assessment of whether reductions in self-reported GI symptom severity are rooted in improvements in gut function. Finally, there was a significant interaction effect for fecal butyrate, and pairwise comparisons revealed significantly reduced butyrate levels in samples from the TOME group compared to Placebo at 4 weeks. Butyrate, a SCFA produced by certain gut bacteria, plays a crucial role in maintaining gut health. It helps reinforce the epithelial defense barrier, regulates the expression of tight junction proteins like Claudin-1, and reduces inflammation by limiting pro-inflammatory cytokines [65]. Thus, while higher levels of fecal butyrate are most often associated with improvements in gut barrier function [65,66,67], some have argued that reduced butyrate levels may indicate increased colonic utilization [68]. Butyrate production is also subject to consumption of fiber in the diet, although we did not observe significant differences in average fiber intake. Since the changes that we observed were only between the two groups at the 4-week timepoint, without significant differences in butyrate levels over time in either group, it is difficult to interpret the significance of these observations.

### 4.2. Mental Health and Gut–Brain Axis

Stress, anxiety, and other aspects of mental health can often trigger or exacerbate GI symptoms [12]. Likewise, disruptions in the gut can impact the gut–brain axis to alter mood and behavior. Therefore, we also explored the impact of TOME supplementation on aspects of mental health, particularly reductions in anxiety and stress, and markers of sympathetic nervous system activation. Although this has not been explored previously with TOME, astaxanthin, a potent antioxidant derived from the microalga *Haematococcus pluvialis*, has been shown to support emotional well-being and cognitive health [69]. In a study involving 28 healthy subjects, participants were supplemented with 12 mg/day of astaxanthin for eight weeks. The results indicated significant improvements in mood, including a 57% reduction in depression scores and a 36% reduction in fatigue scores compared to the placebo. Additionally, there were notable improvements in overall mood (+11%) and vigor (+5%) [69]. Although this study used different assessment tools than the current study, the findings are similar. We showed that there were improvements in perceived stress in both the TOME and Placebo groups, but only the TOME group also showed a significant decrease in Negative Affect Scores from baseline to week 2 and week 4. Interestingly, there was also a significant decrease in Positive Affect Scores in this group between week 2 and week 4, which may indicate individual variability in effects. However, the study population had relatively low levels of stress and anxiety at baseline, so further study in a population specifically recruited for the purpose of establishing mental wellness benefits is warranted. To further support this, plasma levels of chromagranin A were significantly reduced between baseline and week 4 in the TOME group and with TOME compared to Placebo. Plasma chromagranin A has been shown to be elevated in patients with IBD and is associated with regulation of gut barrier function. Muntejewerff et al. [70] showed that intraperitoneal administration of recombinant pancreastatin (PSA), a fragment of chromagranin A, to chromagranin A knockout mice reduced gut barrier function. Chromagranin A is also associated with activation of the sympathetic nervous system and reductions in chromagranin A have been correlated with reduced stress responses. A study of 268 healthy individuals showed that there was a strong positive correlation between chromagranin A in plasma and anxiety and depression scores on the Hospital Anxiety and Depression Scale and subgroups [71]. Chromagranin A is also involved in the regulation of hormone secretion and vascular homeostasis and elevated levels have been associated with CVD [72].

Increased sympathetic nervous system activation is associated with increased risk for hypertension and has other cardiovascular effects [18]. Factors such as chronic stress and anxiety activate the sympathetic nervous system and the HPA axis, thus increasing the risk of developing hypertension through pathophysiological mechanisms such as inflammation [10]. Thus, targeting the gut–brain axis with dietary approaches could have benefits for cardiovascular health. The findings of the current study indicate no major effects of TOME on brachial blood pressure at rest or in response to the CPT, nor for aortic parameters. Though there were differential effects in the responses of some parameters to the CPT, there were no differences between groups, and the findings do not conclusively suggest a benefit or detriment to cardiovascular health. It is important to note that this was a relatively young, healthy population with low cardiovascular risk at baseline. Future studies are needed in populations with increased risk for CVD. Though previous studies have evaluated the influence of diet and nutrition on the gut–brain axis [73], as well as on blood pressure responses to sympathetic nervous system activation (e.g., though CPT) [74], research evaluating the potential interactions of these systems in terms of diet and nutrition are lacking.

### 4.3. Study Limitations and Future Directions

Our compliance data, metabolic panel, and adverse event reports suggest that 4 weeks of TOME supplementation was both safe and tolerable in a population with self-reported mild to moderate GI distress. There was >90% compliance with both groups and none of the attrition was due to increases in symptoms or other study-related adverse events. The major side effect associated with TOME ingestion was increased flatulence, which is commonly associated with the initial adjustment period to gut-targeted supplements and frequently subsides with use [75]. Metabolic panels showed that liver enzymes remained within the reference ranges during the 4-week TOME supplementation period, and we even observed a significant decrease in AST between baseline and week 4. This may be due to compounds in the TOME such as the PUFAs and lutein, which can act as prebiotics and antioxidant compounds [19,60], and which may reduce inflammation in the gut environment to improve digestion, ultimately reducing liver stress.

Some strengths of this study included recruitment of a population with pre-existing GI complaints and the use of both subjective and objective assessments of GI and mental health, as well as the randomized, double-blind, placebo-controlled study design. However, there were several limitations to interpretation of the data. This was a pilot study, so the sample size may not have been powered to detect changes in all measured outcomes, although it was powered to detect significant differences in GI function scores. This is particularly true for some of the molecular markers of gut and immune function, which were highly variable within treatment groups. However, the data presented in this study are important for determining sample sizes for follow-up studies.

In addition, there are a number of variables such as individual variability of responses, background diet and exercise habits, and other genetic and environmental factors that could impact outcomes. A crossover design study would reduce some of the inter-individual variability and allow everyone to be compared back to their baseline and may be more able to detect subtle changes that were masked in this study. As most of the benefits were noted at the week-4 timepoint, it would be interesting to see if the benefits of TOME remain consistent, and if additional changes are observed with a longer study duration. Finally, it will be important to determine whether additional benefits occur in a population with more severe GI dysfunction (i.e., IBS) and/or higher levels of stress and anxiety.

In addition, a better mechanistic understanding of the changes elicited by TOME is needed. This could include supplementing the 16s rRNA sequencing data with more functional measures of the microbiome, including meta-transcriptome, meta-proteome, or metabolome data. It could also include translational studies in animals that would allow evaluation of intestinal physiology and tissue specific changes in specific biomarkers after TOME administration.

## 5. Conclusions

This study found that taking an extract from the microalgae *Tetradesmus obliquus* strain Mi175.B1.a for four weeks was safe and well-tolerated. It reduced symptoms in healthy adults with mild to moderate GI issues. Additionally, the supplement had positive effects on mental health and blood markers of gut function and mental health. While more research is needed to understand these benefits, the results from this pilot study are promising and suggest that this extract could be optimized as a dietary supplement for gut and mental health.

## Figures and Tables

**Figure 1 nutrients-17-00960-f001:**
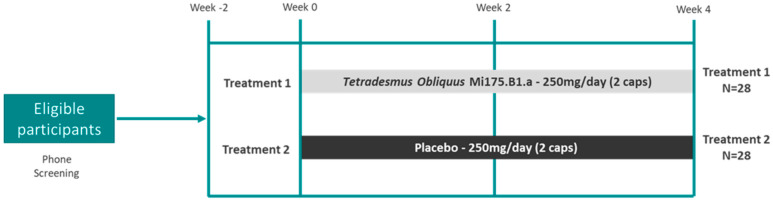
Study design chart. After a phone screening, eligible participants were brought in for a visit to confirm eligibility and provide consent. At the baseline visit, participants were randomized to Treatment 1 (*Tetradesmus obliquus*) or Treatment 2 (Placebo) for 4 weeks. Data were collected at baseline, week 2, and week 4 of the intervention.

**Figure 2 nutrients-17-00960-f002:**
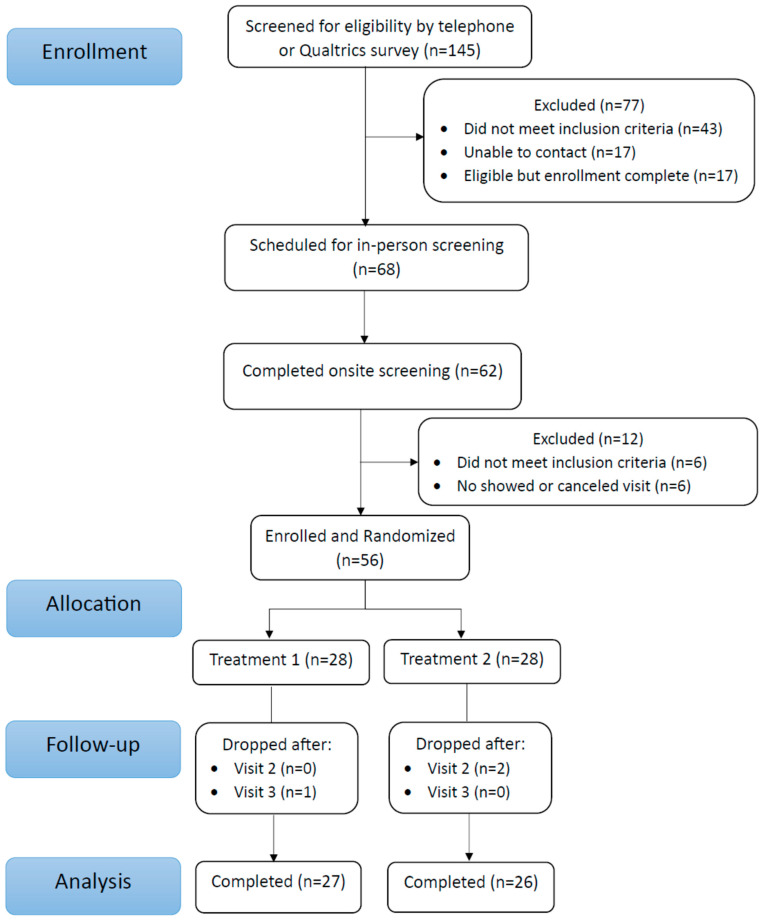
CONSORT Diagram.

**Figure 3 nutrients-17-00960-f003:**
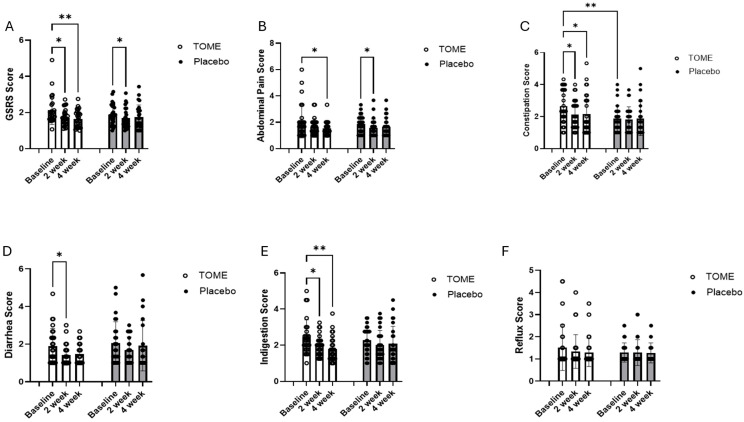
Gastrointestinal symptom severity score as determined by GSRS questionnaire. Average and individual scores are shown at baseline, after 2 weeks, and after 4 weeks for (**A**) GSRS score, (**B**) abdominal pain score, (**C**) constipation score, (**D**) diarrhea score, (**E**) indigestion score, and (**F**) reflux score. * *p* < 0.05, ** *p* < 0.01.

**Figure 4 nutrients-17-00960-f004:**
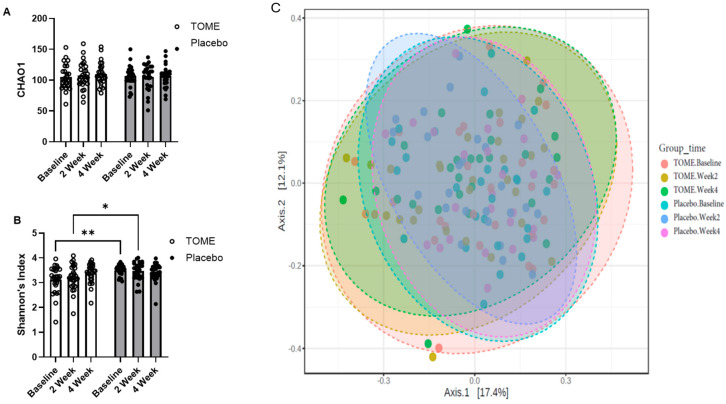
Microbiota was assessed using 16s rRNA sequencing. (**A**) CHAO1 estimates of ASV richness; (**B**) Shannon’s diversity; and (**C**) Bray–Curtis dissimilarity visualized using Principle Coordinates Analysis (PCoA). (* *p* < 0.05, ** *p* < 0.01, data represent unadjusted *p*-values).

**Figure 5 nutrients-17-00960-f005:**
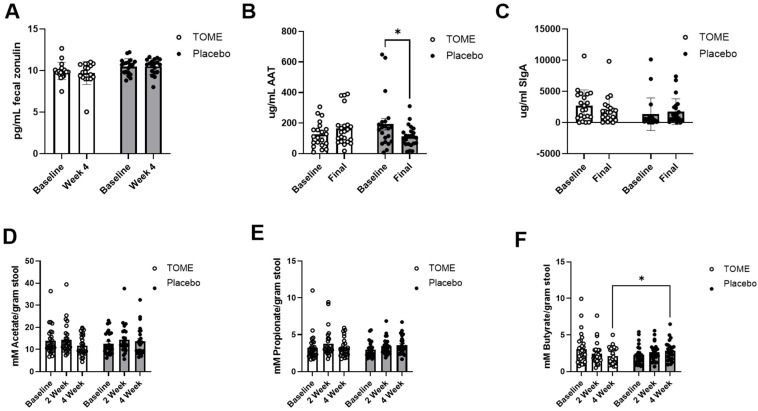
Fecal measures related to gut barrier integrity and microbiota metabolism. Results are shown for (**A**) fecal zonulin, (**B**) fecal alpha-1 antitrypsin (AAT), (**C**) sIgA, and the SCFAs (**D**) acetate, (**E**) propionate, and (**F**) butyrate. * *p* < 0.05.

**Figure 6 nutrients-17-00960-f006:**
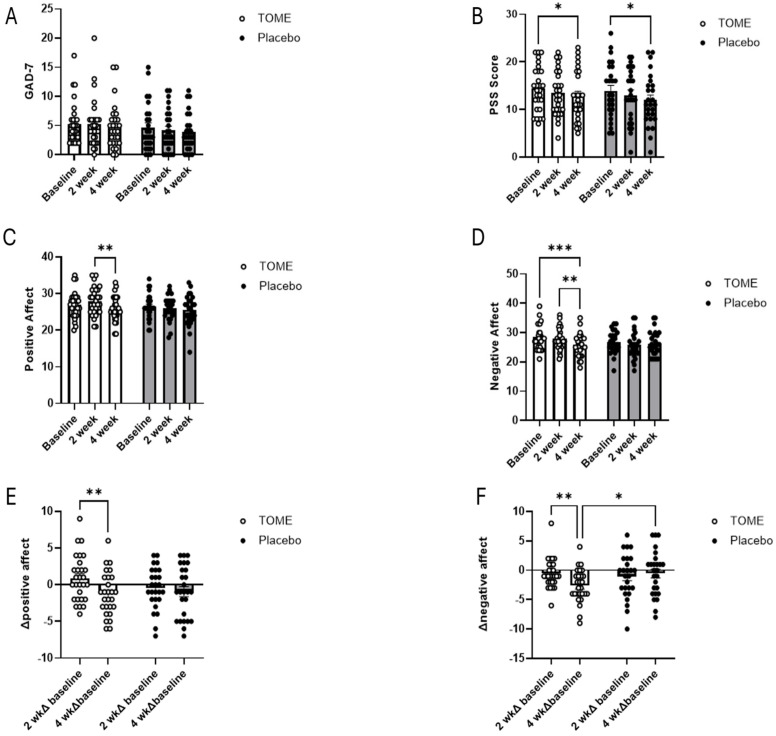
Self-reported psychological measures. (**A**) Generalized Anxiety Disorder-7 (GAD-7), (**B**) Perceived Stress Scale (PSS), (**C**) Positive Affect, (**D**) Negative Affect, (**E**) change in Positive Affect, and (**F**) change in Negative Affect from baseline. (* *p* < 0.05, ** *p* < 0.01, *** *p* < 0.001).

**Figure 7 nutrients-17-00960-f007:**
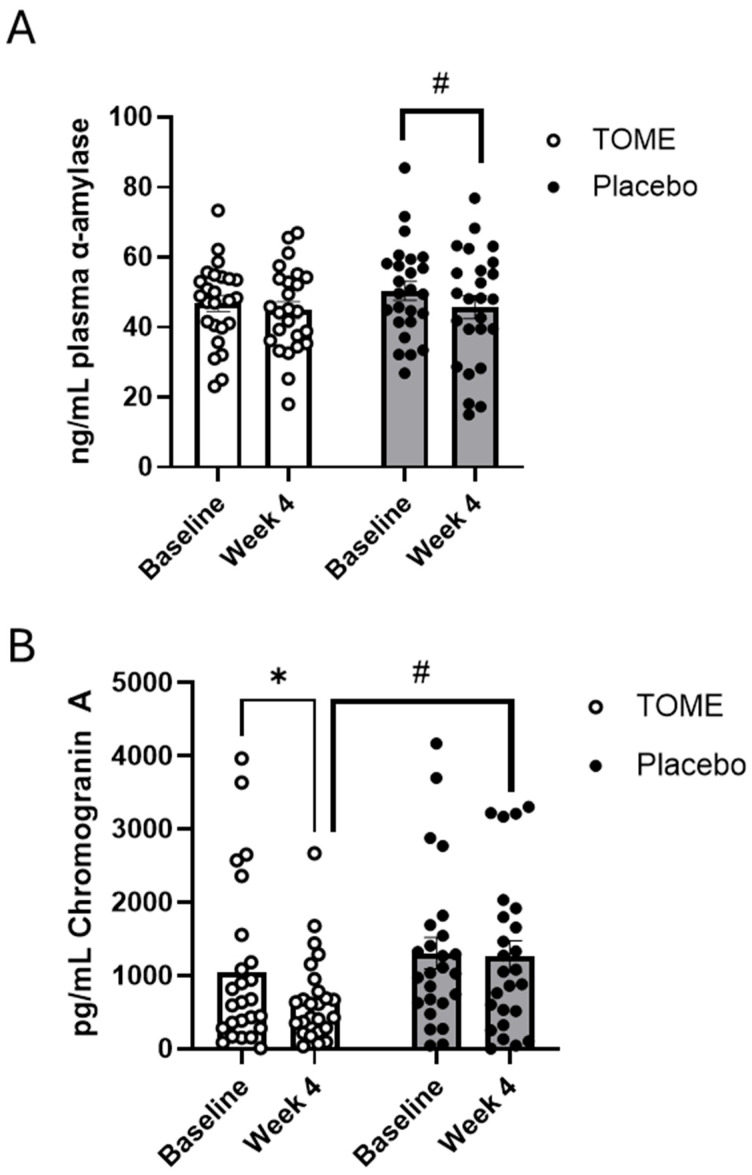
Blood markers of stress. (**A**) plasma a-amylase and (**B**) chromogranin A. (* *p* < 0.05, # *p* = 0.051–0.80).

**Table 1 nutrients-17-00960-t001:** *Tetradesmus obliquus* strain Mi175.B1.a composition.

	Composition	Methods
*Tetradesmus obliquus* Mi175.B1.a	20–35% (*w*/*w*)	
MCT oil based on coconut oil	65–80% (*w*/*w*)	
Mix of non-GMO tocopherols (antioxidant)—E 306	0.45–0.55% (*w*/*w*)	
Total lipids	≥80% (*w*/*w*)	Gravimetry
Proteins	≤5% (*w*/*w*)	Kjeldahl (Titrimetry)
Humidity	<5% (*w*/*w*)	Thermogravimetry
Total PUFAs	3.5–11.5% (*w*/*w*)	GC-FID
Carbohydrates	≤20% (*w*/*w*)	Calculation
Total MUFAs	1–5.5% (*w*/*w*)	GC-FID
Lutein	0.3%(*w*/*w*)	HPLC
Sterol content (g/100 g product)	0.3–0.9	NF en ISO 12228-1 [40]

**Table 2 nutrients-17-00960-t002:** Participant Demographics.

Variable		N	Mean			*p*-Value
Sex F(M)	Placebo	28	16 (12)			0.64
	TOME	28	18 (10)			
Age	Placebo	28	30.1	±	6.25	0.11
	TOME	28	33.7	±	8.68	
	Total	56	31.9	±	7.72	
Height	Placebo	28	170.7	±	9.97	0.94
(cm)	TOME	28	170.5	±	10.03	
	Total	56	170.6	±	10.00	
Body Mass Index (kg/m^2^)	Placebo	28	24.9	±	2.82	0.35
	TOME	28	24.2	±	2.73	
	Total	56	24.6	±	2.77	
Weight	Placebo	28	72.9	±	12.43	0.53
(kg)	TOME	28	70.8	±	12.92	
	Total	56	71.8	±	12.61	
Waist Circumference	Placebo	28	81.5	±	8.43	0.50
(cm)	TOME	28	79.9	±	9.08	
	Total	56	80.7	±	8.72	
Hip Circumference	Placebo	28	101.6	±	7.31	0.16
(cm)	TOME	28	99.1	±	5.63	
	Total	56	100.3	±	6.58	
Resting Heart Rate	Placebo	28	61.4	±	8.75	0.84
(beats/min)	TOME	28	60.9	±	8.66	
	Total	56	61.2	±	8.63	
Systolic Blood Pressure (mmHg)	Placebo	28	113.1	±	10.13	0.34
	TOME	28	110.8	±	7.10	
	Total	56	112.0	±	8.72	
Diastolic Blood Pressure	Placebo	28	67.8	±	8.05	0.85
(mmHg)	TOME	28	67.4	±	6.27	
	Total	56	67.6	±	7.13	

Data are presented as means ± standard deviations.

## Data Availability

Sequence data are publicly available from QIITA (https://qiita.ucsd.edu/) under study ID 15884.

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
