# Peer review of "Effects of Supplementation with Microalgae Extract from Tetradesmus obliquus Strain Mi175.B1.a on Gastrointestinal Symptoms and Mental Health in Healthy Adults: A Pilot Randomized, Double-Blind, Placebo-Controlled, Parallel-Arm Trial"

_nutrients, 2025, doi:10.3390/nu17060960_

Round 1

Reviewer 1 Report

Comments and Suggestions for Authors

It is an interesting study on the effect of a microalgae on gastreointestinal symptoms and mental health in healthy adults

The study is well designed; it is a pilot study. Thus, unfortunately without power analysis, leading to no significant differences in many parameters between the two groups.

I would like authors to write a paragraph in discussion explaning why the were based on other studies having sample sizes of 20-30 participants [only!] which led them to present questionable results. In other words, why I have to say 'ok, they have not enough enthousiastic results, but it is a clever idea, lets accepted for publication"

Additionally, I would like to read a paragraph in Introduction about microalgae and specifically about the Mi175.B1.a

Author Response

  1. The introduction should include a more detailed discussion of the gut microbiome’s role in mental health. There is little elaboration on how specific microbial changes, such as shifts in bacterial composition or metabolite production, may influence mood, anxiety, and stress regulation.

We have now included a more detailed description of some of the processes by which the gut microbiota can influence neuroinflammation and impact both cardiovascular function and mental health. We did avoid any mention of specific microbial taxa as the GBA studies are still generally associative rather than causal. Furthermore, we did not provide a specific microbe or observe any significant taxa shifts in our study that would detailed discussion of specific taxa impacts on GBA.

  1. The introduction could benefit from a broader comparison of different microalgae species and their bioactive components. The current version briefly mentions other strains, but it does not provide a clear differentiation between Tetradesmus obliquus and other microalgae that have been studied for similar effects.

There is still fairly limited information regarding microalgae as they relate to human health; however, we have expanded this section somewhat and mentioned some of the known compositional differences between T. obliquus and other microalgae. To date, the majority of studies with T. obliquus have focused on its production methodology; therefore, this contributes to the novelty of our work.

  1. It would be useful to discuss potential limitations of dietary interventions for gut and mental health. The article does not acknowledge challenges such as individual variability in gut microbiota composition, dietary habits, or lifestyle factors that may influence the efficacy of supplementation.

We have expanded on the discussion to include more of the study limitations as suggested by the reviewer.

  1. Also, IMO, the article’s conclusions should be written in a more popular science language so that individuals unfamiliar with scientific terminology can also understand the key findings.

We have rewritten the entire conclusion to use more plain language.

Reviewer 2 Report

Comments and Suggestions for Authors

This article presents a well-conducted study exploring the effects of microalgae extract from Tetradesmus obliquus strain Mi175.B1.a on gastrointestinal health and mental well-being. However, while the study itself is strong, the introduction feels somewhat incomplete. There are a few key aspects that could be expanded to strengthen the overall narrative:

1. the introduction should include a more detailed discussion of the gut microbiome’s role in mental health. There is little elaboration on how specific microbial changes, such as shifts in bacterial composition or metabolite production, may influence mood, anxiety, and stress regulation. 

2. the introduction could benefit from a broader comparison of different microalgae species and their bioactive components. The current version briefly mentions other strains, but it does not provide a clear differentiation between Tetradesmus obliquus and other microalgae that have been studied for similar effects. 

3. it would be useful to discuss potential limitations of dietary interventions for gut and mental health. The article does not acknowledge challenges such as individual variability in gut microbiota composition, dietary habits, or lifestyle factors that may influence the efficacy of supplementation. 

Also, IMO, the article’s conclusions should be written in a more popular science language so that individuals unfamiliar with scientific terminology can also understand the key findings.

Author Response

  1. It is an interesting study on the effect of a microalgae on gastreointestinal symptoms and mental health in healthy adults. The study is well designed; it is a pilot study. Thus, unfortunately without power analysis, leading to no significant differences in many parameters between the two groups. I would like authors to write a paragraph in discussion explaning why the were based on other studies having sample sizes of 20-30 participants [only!] which led them to present questionable results. In other words, why I have to say 'ok, they have not enough enthousiastic results, but it is a clever idea, lets accepted for publication".

We have added to the discussion the limitation of our sample size and highlighted the utility of this study which is that we now have sufficient data for a variety of outcomes to calculated well powered sample sizes to be used in the design of future trials.

  1. Additionally, I would like to read a paragraph in Introduction about microalgae and specifically about the Mi175.B1.a

There is still fairly limited information regarding microalgae as they relate to human health; however, we have expanded this section somewhat and mentioned some of the known compositional differences between T. obliquus and other microalgae. To date, the majority of studies with T. obliquus have focused on its production methodology; therefore, this contributes to the novelty of our work.

Round 2

Reviewer 1 Report

Comments and Suggestions for Authors

The revision made by authors is OK, to my part

thank you